# Development and Validation of Benzophenone Derivatives in Packaged Cereal-Based Foods by Solid–Liquid Extraction and Ultrahigh-Performance Liquid Chromatography–Tandem Mass Spectrometry

**DOI:** 10.3390/foods11091362

**Published:** 2022-05-07

**Authors:** Yu-Fang Huang, Jun-Jie Huang, Xuan-Rui Liu

**Affiliations:** 1Department of Safety, Health and Environmental Engineering, National United University, Miaoli 36063, Taiwan; k10220202@gmail.com (J.-J.H.); aass1aass35@gmail.com (X.-R.L.); 2Center for Chemical Hazards and Environmental Health Risk Research, National United University, Miaoli 36063, Taiwan; 3Institute of Food Safety and Health Risk Assessment, National Yang Ming Chiao Tung University, Taipei 11221, Taiwan

**Keywords:** benzophenone derivatives, packaged cereal-based foods, solid–liquid extraction, ultra-high-performance liquid chromatography–tandem mass spectrometry

## Abstract

We established and validated a sensitive multi-residue analytical method for identifying benzophenone (BP) and nine BP derivatives (2,4-dihydroxybenzophenone [BP-1], 2,2′,4,4′-tetrahydroxydroxybenzophenone, 2-hydroxy-4-methoxy benzophenone, 2,2′-dihydroxy 4-methoxy benzophenone, 2-hydroxybenzophenone [2-OHBP], 4-hydroxybenzophenone, 4-methylbenzophenone [4-MBP], methyl-2-benzoylbenzoate, and 4-benzoylbiphenyl). Solid–liquid extraction pretreatment and ultra-high-performance liquid chromatography–tandem mass spectrometry (UHPLC–MS/MS) were employed in an analysis of 85 packaged cereal-based food samples (25 pastry, 50 rice, and 10 noodle samples). The method had satisfactory linearity (R^2^ ≥ 0.995), low limits of detection (pastry: 0.02–4.2 ng/g; rice and noodle: 0.02–2 ng/g), and favorable precision, with within-run and between-run coefficient of variation ranges of 1–29% and 1–28%, respectively. BP and 4-MBP were detected in 100% of the pastry samples, and BP-1 and 2-OHBP were found in 76% and 56% of the pastry samples, respectively. BP and 2-OHBP were found in 92% and 38% of the rice samples, respectively. BP was found in 50% of the noodle samples. BP contributed the most to the total level of BPs in pastries, with significantly higher mean ± standard deviation (range) levels for pastries (26.8 ± 32.6 [1.8–115.4] ng/g) than rice (1.2 ± 2.0 [0.4–13.4] ng/g) and noodles (0.7 ± 0.7 [0.4–1.9] ng/g); *p* < 0.0001). The trace levels of 4-MBP identified in the samples demonstrate the need for the development of analytical methods with high sensitivity and specificity; the proposed method satisfies this need.

## 1. Introduction

Cereal-based products, such as rice and noodles, are staple foods in many countries; they comprise approximately 75% of the average individual’s carbohydrate intake (mainly in the form of starch) and 6–15% of their protein, vitamin, and mineral intake [1]. Studies have investigated the importance of cereals-based products, particularly wholegrain products, in providing these carbohydrates due to the presence of dietary fiber and bioactive compounds in such products. Diets rich in dietary fiber and whole grains are associated with a decreased risk of atherosclerotic cardiovascular diseases [2,3], type 2 diabetes [3], and obesity [2]. In the packaging of these cereal-based products, benzophenone (BP) and its derivatives (BPs) are often added to the plastic packaging as ultraviolet (UV) blockers and are used as photoinitiators (PIs) in UV-curable inks for printed food-packaging materials [4,5]. BPs are widely used for various applications, including pharmaceuticals, insecticides, and personal care products [4,5]. However, BP, 4-methylbenzophenone (4-MBP), 4-benzoylbiphenyl (PBZ), and methyl-2-benzoylbenzoate (M2BB) have a strong tendency to migrate into food and cause contamination [6]. Their widespread application therefore results in human exposure to BP and BPs through dermal absorption [7], inhalation [8], and ingestion due to migration from food-contact materials [9,10]. In 2009, German authorities reported a high migration rate of PIs (i.e., 4-MBP) of up to 798 ng/g from cardboard packaging materials into crunchy muesli, identified using the Rapid Alert System for Food and Feed [11]. The European Union has regulated regulations limiting the migration of BP in food to 0.6 mg/kg [12].

BPs belong to a group of endocrine-disrupting chemicals that can interfere with hormone action [13,14] and 2-hydroxy-4-methoxy benzophenone (BP-3) that can cause breast cancer and endometriosis [15]. Epidemiological studies have reported that prenatal urinary 2,4-dihydroxybenzophenone (BP-1) and BP-3 may lead to decreased gestation age and fetal growth, especially in girls [16,17]. According to the International Agency for Research on Cancer and the American National Toxicology Program, sufficient evidence has been presented to support BP carcinogenicity in mice and rats; these mice and rats have been reported to present with hepatocellular and renal tubule adenomas, indicating carcinogenicity in humans (Group 2B) [5,18].

Because of these risks associated with the potential adverse effects of BPs in humans and the ubiquitous application of BPs-containing products, a reliable analytical technique for quantifying BP levels in foods is crucial. BP content in cereal-based products can be quantified through gas chromatography (GC) combined with mass spectrometry (MS) [10,19,20,21] and through high-performance liquid chromatography (HPLC) [22,23] and ultra-performance liquid chromatography (UPLC) [24,25,26] combined with tandem MS (MS/MS). Some pretreatment techniques have been proposed for identifying the presence of BPs in cereals and cereal-based foods; among these techniques, solid–liquid extraction (SLE) with acetonitrile (ACN) [10,19,20,23,24] and dichloromethane [19,20] and pressurized liquid extraction (PLE) [21] are mainly used. In addition, fast pesticide extraction (FaPEx) and quick, easy, cheap, effective, rugged, and safe (QuEChERS) approaches are used to identify BPs in breakfast cereals [27] and fish [28]. The QuEChERS method comprises extraction with ACN and dispersive solid-phase extraction (d-SPE) aliquot purified with MgSO_4_ and SPE sorbents, such as primary secondary amine (PSA) and C18, to remove pigments, lipids, and fatty acids. FaPEx is a simplified and fast design of the QuEChERS technique; it can be used to extract multipesticide residues. FaPEx involves single-use, prefilled, sealed SPE cartridges with PSA, C18, and graphitized carbon black sorbents (GCB) [29]. An approach featuring the SLE method followed by SPE with a hydrophilic–lipophilic balance (HLB) cartridge was proposed in a previous study [25]. However, some methods are limited by the number of BPs that can be simultaneously quantified and by instrument sensitivity, which has a detection limit of 10–150 ng/g [10,19,23,24,26].

This study develops an SLE technique combined with ultra HPLC (UHPLC)–MS/MS for a simultaneous analysis of the presence of BP and nine BPs (BP-1,2,2′,4,4′-tetrahydroxydroxybenzophenone [BP-2], BP-3,2,2′-dihydroxy 4-methoxy benzophenone [BP-8], 2-hydroxybenzophenone [2-OHBP], 4-hydroxybenzophenone [4-OHBP], 4-MBP, M2BB, and PBZ). A stable isotope labeling (SIL)-assisted technique using ^13^C-labeled or d-labeled internal standards (ISs) for UHPLC–MS/MS are applied to achieve precise quantification and mitigate measurement uncertainty. The SIL-UHPLC–MS/MS method was applied to commercial packaged cereal-based samples from Taiwan to validate its feasibility.

## 2. Materials and Methods

### 2.1. Reagents and Chemicals

The chemicals were of analytical grade and had certificates of analysis. PBZ was obtained from Alfa Aesar (Lancashire, UK); BP-1 and BP-3 were acquired from AccuStandard (New Haven, CT, USA); BP, 2-OHBP, 4-OHBP, M2BB, and 4-MBP were obtained from Sigma-Aldrich (St. Louis, MO, USA), and BP-2 and BP-8 were supplied by Tokyo Chemical Industry (Tokyo, Japan). SIL-IS, BP-d_5_, and BP-3-d_5_ were purchased from Sigma-Aldrich (Burlington, MA, USA); BP-1-d_5_, 4-MBP-d_3_, BP-8-d_3_, diOHBP-^13^C_6_, and 4-OHBP-d_4_ were obtained from Toronto Research (North York, Toronto, Canada). All standards and SIL-IS had purities of >97%. Anhydrous magnesium sulfate (≥99%, MgSO_4_), LC-grade ACN, formic acid (≥88%), acetic acid (≥99.7%), and LC–MS-grade methanol (MeOH) were purchased from J.T. Baker (Phillipsburg, NJ, USA). Sodium chloride (>99%, NaCl) was obtained from PanReac (Castellar del Vallès, Barcelona, Spain). The bulk sorbent, Sepra C18-E and PSA, and Strata C18-T SPE cartridges (1 mL, 100 mg) were obtained from Phenomenex (Torrance, CA, USA) and Sigma-Aldrich. FaPEx-cer was obtained from Silicycle (Quebec, QC, Canada). Oasis PRiME HLB cartridges (1 mL, 30 mg) were obtained from Waters (Milford, MA, USA). The Milli-Q water of the study was produced by a Sartorius Ultrapure water system to reach a resistivity of 18.2 MΩ cm (Savska, Zaprešić, Croatia).

### 2.2. Packaged Sample Collection and Preparation

A total of 85 packaged cereal-based foodstuffs were randomly selected from bakeries and supermarkets in Taiwan. All the samples were domestic and were classified into three groups: pastries (*n* = 25), rice (*n* = 50), and noodles (*n* = 10). The collected foods were popular brands in Taiwan. The types of packaging materials used on the internal side of the products’ packaging were examined, and the food contact materials were recorded to be polypropylene (PP) and polyethylene terephthalate (PET) plastics. PP was the food contact material in 19 pastry, 16 rice, and 1 noodle samples, and PET was the food contact material in 6 pastry, 0 rice, and 8 noodle samples. The types of packaging materials were unknown in 34 rice and 1 noodle samples. Because no blank matrices with an absence of BPs were available, unwrapped samples of pastry (*n* = 6), rice (*n* = 6), and noodles (*n* = 6) were analyzed through UHPLC–MS/MS; among them, three samples of each type were mixed and selected as three types of blank matrices because the BPs were below the limit of detection (LOD).

### 2.3. Preparation of Standards

A stock standard solution of each analyte (500 mg/L) was individually prepared in ACN. The standard working solutions of 50 mg/L, which were prepared by dilution with MeOH in an appropriate volume of the solution, were used to spike the calibration curves of solvents and cereal-based matrices. BP-d_5_, BP-1-d_5_, 4-MBP-d_3_, BP-8-d_3_, diOHBP-^13^C_6_, 4-OHBP-d_4_, and BP-3-d_5_ were used as SIL-IS for BP, BP-1, 4-MBP, BP-8, 2-OHBP, 4-OHBP, and BP-3, respectively. Because no SIL-IS was available for BP-2, M2BB, and PBZ, 4-OHBP-d_4_ was used as an SIL-IS for BP-2 and M2BB, and BP-3-d_5_ was used as an SIL-IS for PBZ. The SIL-IS mixture was diluted to 20 μg/L.

### 2.4. Analysis of UHPLC–MS/MS Method

With the exception of BP-2, BPs were detected using a UHPLC–MS/MS system (Shimadzu 8045, Kyoto, Japan) connected to a triple-quadrupole MS system with an electrospray ionization positive mode. The analytical column was a UPLC Waters BEH C18 column (1.7 μm, 2.1 mm × 100 mm) and had flow rate of 0.3 mL/min. Mobile phase A was MeOH containing 0.1% formic acid, and mobile phase B was deionized water. The elution gradient was 80–20% B at 3.5 min, 20% B at 1 min, 20–10% B at 1 min, 10% B at 4 min, and 10–80% B at 0.1 min, and re-equilibrated at 80% for 3.9 min. The total run time was 13.5 min, and the sample injection volume was 10 µL. BPs were monitored under multiple reaction monitoring (MRM) modes. The interface, desolvation line, and heat block temperatures were 300, 240, and 400 °C, respectively. The flow rates of the nebulizing, heating, and drying gases were 3, 10, and 10 L/min, respectively. The tandem MS parameters, ion transitions for quantification and qualifications, retention time, and collision energy are presented in Table 1. Data acquisition and processing were conducted using LabSolution (version 5.93, Shimadzu, Kyoto, Japan).

### 2.5. Pretreatment Approaches

Six procedures, namely SLE, FaPEx, QuEChERS with and without cleanup, and the SPE-based method with HLB and C18 cartridges, were compared to select a suitable sample pretreatment method and were tested with five replicates.

Method A, SLE: A homogenized cereal-based food sample (0.5 g) was loaded into a 12 mL glassware tube, and a standard (STD) and SIL-IS (20 and 8 ng/g) and ACN (5 mL) with 1% acetic acid were added. The mixtures were shaken for 1 min, allowed to stand for 60 min, and centrifuged at 2000× *g* for 10 min. Subsequently, the supernatant solution was dried by nitrogen gas. Finally, the residue was dissolved in 200 μL of MeOH and filtered using 0.22 μm polytetrafluoroethylene filters.

Method B, FaPEx: A homogenized sample (0.5 g) was loaded into a 12 mL glassware tube. Subsequently, pure water (1 mL) was added, and the sample was spiked with an STD and SIL-IS and vortexed for 1 min, and we then waited for 30 min. ACN (5 mL) with 1% acetic acid was added; the mixtures were vortexed for 30 s and centrifuged at 2000× *g* for 10 min. Thereafter, the supernatant was transferred to a FaPEx-cer kit with MgSO_4_, PSA, C18, and GCB, at a liquid flow rate of 1 drop/s and dried by nitrogen gas. Finally, the residue was treated in the same manner as that used in the SLE method.

Methods C–D, QuEChERS with and without a cleanup: The original QuEChERS method comprises several steps: (1) addition of an IS; (2) extraction using ACN; (3) partitioning with MgSO_4_ and NaCl; (4) dispersive SPE aliquot purified with MgSO_4_ and SPE sorbents; and (5) adjustment of the pH [30]. The QuEChERS method employed in this study was described in a previous study [27]. A homogenized sample (0.5 g) was placed in a 12 mL glassware tube, and an STD and SIL-IS and deionized water (1 mL) were added. The mixtures were shaken for 1 min and allowed to stand for 60 min, and ACN (5 mL) with 1% acetic acid was added. The mixtures were shaken for 1 min, extraction salt packages (4 g of anhydrous MgSO_4_ and 1 g of NaCl) were added, and the mixtures were vigorously shaken again for 1 min. For samples without a cleanup procedure, the extract was centrifuged for 10 min at 2000× *g*. For those with a cleanup procedure, a supernatant was added to the cleanup sorbents, which contained 0.24 g of MgSO_4_, 0.24 g of C18-E, and 0.08 g of PSA. Finally, the residue was treated in the same manner as that in the SLE procedure.

Methods E–F, SPE-C18, and SPE-HLB: A homogenized sample (0.5 g) was loaded into a 12 mL glassware tube, and an STD, SIL-IS, and ACN (5 mL) with 1% acetic acid were added. The mixtures were shaken for 1 min and centrifuged at 2000× *g* for 10 min. Thereafter, the supernatant was prepared for purification through SPE by using Strata C18-T and Oasis PRiME HLB. First, the SPE cartridges were conditioned using 2 mL of each MeOH and deionized water in sequence. Subsequently, the supernatant was loaded onto the cartridges. The prepared solutions (no washing solvents, water, and 10% MeOH) were examined with 2 mL for washing efficiency, and MeOH and ACN were tested with 2 mL for elution efficiency. Finally, the eluent was collected and concentrated to dryness by nitrogen gas per the aforementioned SLE procedure.

### 2.6. Method Validation

The method was validated according to the guidelines established in the United States [31] for linearity, the matrix effect (ME), the LOD, the limit of quantification (LOQ), recovery, and precision. Blank matrix samples were unwrapped samples of pastries, rice, and noodles, which were analyzed through UHPLC–MS/MS and had BP contents below the LOD. Linearity was evaluated using solvent-matched and matrix-matched calibration standards with 8 levels (0.04, 2, 10, 30, 60, 72, 92, and 100 ng/g, with an SIL-IS of 8 ng/g) for pastries and 10 levels (0.04, 0.4, 0.6, 1.2, 2.4, 4, 6, 12, 24, and 40 ng/g, with an SIL-IS of 8 ng/g) for rice and noodles. The calibration curves of solvents and matrices were obtained by plotting the quotients of the peak areas of BPs and their corresponding SIL-IS against the levels of the standards. The MEs were evaluated through the comparison of the slopes of standards in solvents with matrix-matched standards. The LOD and LOQ were defined as the levels with signal-to-noise ratios of ≥3 and ≥10, respectively. Blank pastry samples with 3 spiking levels (4, 40, and 80 ng/g) and rice and noodle samples with 3 levels (0.4, 4, and 40 ng/g) were used to evaluate the recovery and precision of the method. The within-run and between-run recoveries were evaluated on the basis of the mean recovery for these spiked samples on the same day (*n* = 5) and over 3 consecutive days (*n* = 15), respectively. The precision was used to determine the relative standard deviation (RSD); the within-run and between-run precisions were assessed on the basis of the standard deviation (SD) of the recovery percentage of the spiked samples on a given day (*n* = 5) and 3 consecutive days (*n* =15). The validation criteria of the Codex Alimentarius were 70–120% for ME and recovery and RSD < 20% for precision.

To evaluate the potential procedure-associated contamination and verify the instruments’ performance, a solvent blank, procedural blank, and medium-spiked sample were applied at intervals of every 10 samples during sample analysis.

### 2.7. Statistical Analysis

The experimental results are presented in terms of the means (SDs, range). Mann–Whitney U test and Kruskal–Wallis tests were used to evaluate the differences in BPs in three types of cereal-based foods and their packaging materials. Statistical analysis was performed using SPSS (version 19.0, SPSS, Chicago, IL, USA), and the significance level was *p* < 0.05.

## 3. Results and Discussion

### 3.1. Selection of Sample Pretreatment Method

Many studies have investigated the migration of UV ink to foodstuffs [32,33,34]; however, few have simultaneously investigated BPs in cereals and cereal-based foods. Bugey et al. (2013) [21] reported a pretreatment method involving PLE and analysis performed through GC–MS, which yielded an LOQ of 60 ng/g for BP in cereal-based foods. Anderson et al. [19] and Ibarra et al. [10] reported a technique featuring SLE followed by GC–MS for quantifying BP in packaged foods (e.g., burgers, cakes, and rice). The LOD and LOQ of BP were 10 ng/mL and 50–250 ng/mL, respectively. Furthermore, Jung et al. [23] and Van Den Houwe et al. [24] reported an SLE method, with HPLC–MS/MS applied for the quantification of BP, 4-MBP, and 4-OHBP in cereal-based foods with LODs of 31–38, 3–13, and 3 ng/g, respectively. Van Den Houwe et al. [25] performed SLE followed by SPE with an HLB cartridge and then applied the UPLC–MS/MS method to analyze BP, 4-MBP, and 4-OHBP in dry foods, with LODs ranging from 0.1 to 4 ng/g. Chang et al. [26] applied the QuEChERS technique without a cleanup procedure and then used UPLC–MS/MS to detect PIs in breakfast cereals, with LOQs of 20, 10, and 40 ng/g for BP, 4-MBP, and 2-OHBP, respectively. Gallart-Ayala et al. [22] used the QuEChERS technique followed by HPLC–MS/MS for the analysis of 11 PIs in packaged food, which yielded the LOQs of 2.3 and 0.7 ng/g for BP and PBZ, respectively. However, the aforementioned methods were limited to the few BPs that can be simultaneously analyzed and that have a high LOD. Huang et al. [27] developed a FaPEx method, with UHPLC–MS/MS used to analyze BPs in cereals, with LODs ranging from 0.001 to 0.3 ng/g. UHPLC achieves a rapid chromatographic technique with a higher resolution, and lower mobile phase volume compared to HPLC.

In this study, we applied the SLE, FaPEx, QuEChERS with or without a cleanup, SPE-HLB, and SPE-C18 methods in spiked blank samples (pastries, rice, and noodles) at standards and SIL-IS level of 20 and 8 ng/g, respectively. A comparison of the six methods for the extraction and cleanup of the BPs revealed that the recoveries were in the order of SLE > FaPEx > QuEChERS without cleanup ≈ QuEChERS with cleanup > SPE-C18 > SPE-HLB among three matrices (Figure 1). To optimize the SPE procedures, two SPE cartridges, Oasis PRiME HLB and Strata C18, were used to investigate washing and elution efficiencies. In our experiment, 2 mL of water and 10% MeOH were added in the washing step, and 2 mL of MeOH and ACN were added in the elution step. The results are displayed in Appendix A. The SPE-C18 had a higher recovery, although all wash solvents minimized the recovery. The MeOH elution solvent achieved the best recovery. Therefore, the optimization of SPE-C18 was proposed to eliminate the need for a wash procedure and elution with MeOH. In addition, as presented in Appendix A, the blank matrix among the six methods indicated BPs below the LOD, with the exception of 2-OHBP and BP. 2-OHBP was detected using the QuEChERS, and BP was mainly identified using the FaPEx and QuEChERS. Because of its high recovery and lack of background contamination, the SLE approach was proposed for BPs analysis in cereal-based foods. A flowchart of sample pretreatment is listed in Figure 2. Figure 3 presents the chromatograms of a solvent sample spiked with 80 ng/g of BP and BP analog standards.

### 3.2. Method Validation

The efficacy of the developed method was validated in accordance with the of the Codex Alimentarius of the United States [31]. Because the analyte contents in the pastry samples were higher than those in the rice and noodles samples, for the pastries, we used a wider working range in validating the recovery and precision of the method, with two 5-point standard calibration curves covering the range of 0.04–60 ng/g constructed for 4 and 40 ng/g and a curve covering the range of 60–100 ng/g constructed for 80 ng/g. For the rice and noodle samples, two 5-point standard calibration curves covering the range of 0.04–4 ng/g were constructed for 0.4 ng/g and a curve covering the range of 4–40 ng/g was constructed for 4 and 40 ng/g.

According to the solvent and matrix-matched calibration curves, all analytes exhibited good linearity (R^2^ > 0.995) in the range of 4–40 ng/g (Table 2 and Table 3). The MEs of BPs among the pastry, rice, and noodle samples were 73–157%, 55–106%, and 26–138%, respectively, indicating that the MEs were acceptable for BP and BP-1. Consequently, the matrix-matched standard solutions were selected for calibration. The LODs and LOQs of BPs among pastry, rice, and noodle samples were 0.08–1.3 and 0.02–4.2, 0.01–1.0 and 0.02–2.0, and 0.01–1.0 and 0.02–2.0 ng/g, respectively. With regard to the recoveries and precision in within-run and between-run tests at 3 spiking levels, the mean within-run and between-run recoveries of the BPs among the pastry, rice, and noodle samples were 45–148% and 44–150%, 75–125% and 70–128%, and 75–145% and 73–152% (Table 4). The mean within-run and between-run precisions (%RSD) of BPs among the 3 matrices were 1–11% and 3–18%, 1–29% and 1–22%, and 1–26% and 1–28%, respectively (Table 5). Most of the BPs satisfied the validation criteria, with the exception of BP-1 and PBZ [31].

As presented in Table 6, PLE and SLE were the most commonly used pretreatment techniques for extracting BPs from packaged cereal-based foods [10,19,21,23,24]. The LODs, LOQs, and recoveries of these methods ranged from 10–38 ng/g, 50–250 ng/g, and 85–115%, respectively, for the extraction of BP; the LODs ranged from 3–13 ng/g and were 3 ng/g with recoveries ranged from 86% to 92% in the extraction of 4-MBP and 4-OHBP, respectively. d-SPE, such as SPE, QuEChERS, and FaPEx, techniques have been applied for the extraction of BPs from packaged dry foods and cereals [22,25,26]. The LOQs for the extraction of BP, 4-MBP, 2-OHBP, and 4-OHBP were 10–20, 10, 40, and 0.6 ng/g, respectively, with recoveries ranging from 84% to 123%. We previously developed an FaPEx method and used UHPLC–MS/MS to analyze BPs in oatmeal and corn flakes with LODs and recoveries ranging from 0.001 to 0.3 ng/g and 79% to 121%, respectively [27].

Thus, we developed a more efficient and simpler approach that achieved satisfactory results with a lower LOD and higher precision for detection of BPs in the samples of pastries, rice, and noodles. Moreover, the application of SIL-IS in this approach achieved highly accurate quantification by compensating for recoveries and MEs and by reducing measurement uncertainty.

### 3.3. Applications of Samples of Popular Food Products in the Taiwanese Market

To verify the suitability of the validated method, 85 samples were subjected to analysis. The detection rates and mean (SD, range) levels of BPs in the samples of the pastry (*n* = 25), rice (*n* = 50), and noodle (*n* = 10) samples are presented in Table 7. Of the 10 analytes, 6 were detected within the range of 2–100% among the 3 cereal-based foods, with BP-2, BP-8, M2BB, and PBZ being below the LOD.

In the pastry samples, six analytes were identified within the range of 4–100%, with BP and 4-MBP detected in 100% of the samples and BP-1, 2-OHBP, BP-3, and 4-OHBP detected in 76%, 56%, 32%, and 4% of the samples, respectively. In the rice samples, four analytes were identified within the range of 2–92%, with BP, 2-OHBP, 4-MBP, and BP-1 detected in 92%, 38%, 16%, and 2% of the samples, respectively. In the noodle samples, BP and 4-OHBP were detected in 50% and 10% of the samples, respectively. These results suggest that BP, BP-1, 2-OHBP, and 4-MBP were prevalent in pastries, and BP was ubiquitous in rice and noodles, which is in line with the findings reported in Belgium [25], Germany [23], Italy [34], Spain [22], Switzerland [21], the United Kingdom [19,20], and Taiwan [27]. BP contributed the most to the total level of BPs in the pastry samples; the second highest contributor was BP-3, with mean ± SD (range) levels of 21.6 ± 37.6 [0.4–85.3] ng/g. BP levels were significantly higher in the pastry samples (26.8 ± 32.6 [1.8–115.4] ng/g) than they were in the rice (1.2 ± 2.0 [0.4–13.4] ng/g) and noodle (0.7 ± 0.7 [0.4–1.9] ng/g); *p* < 0.0001) samples. 2-OHBP levels were significantly higher in the pastry (6.9 ± 6.4 [0.7–23.0] ng/g) than in rice (0.8 ± 3.6 [0.5–25.3] ng/g; *p* < 0.0001) samples. 4-MBP levels were similarly significantly higher in the pastry (5.1 ± 4.2 [0.5–14.4] ng/g) than in rice (0.1 ± 0.2 [0.4–0.9] ng/g; *p* = 0.0001) samples. The higher levels of BP, 2-OHBP, and 4-MBP in pastry samples than in rice and noodle samples may be attributed to the fat content of food products having a strong influence on the migration process [33]. The BP and 4-MBP levels found in this study agree with those reported in Belgium [24] and Germany [23], which were 4 and 13 ng/g. The range of BP levels found in this study was slightly higher than those in samples of cakes, pastries, rice, and noodles reported in Belgium [25] and Spain [10], which were <4–20 and <10–54 ng/g, respectively, but lower than those reported in Germany and the United Kingdom, which were 15–1559 [23], <LOD–439 [20], and 180–2000 ng/g [19]. The high levels of BP in cereal-based foods may be attributed to food contamination from packaging, printed cardboard packaging material, or additional plastic wrapping [10,20,23,32]. In this study, 19 and 6 pastry samples were packaged with PP and PET plastics, respectively, and no significant difference in the mean BPs (BP, BP-1, and 4-MBP) levels was observed among the packaging materials (all *p* > 0.05). However, studies with larger sample sizes are warranted to investigate the association between packaging materials and BPs.

## 4. Conclusions

We developed and validated the FaPEx method for the oatmeal and corn flakes [27] and rice cereal [35] samples. This is the first study in which a simple and efficient SLE UHPLC–MS/MS method was simultaneously applied for the identification of BP and nine BPs in samples of pastry, rice, and noodle. The method exhibited excellent results with the advantages of low background contamination. More specifically, LODs were below the ng/g level, and analyses of BP and BPs had high recovery and precision in the analyses of cereal-based foodstuffs. The results indicate that BP, BP-1, 2-OHBP, and 4-MBP re the most abundant analytes in pastries. The observed trace levels of 4-MBP in the samples indicate a need for the development of analytical methods with high sensitivity and specificity; the proposed method fulfills this need.

## Figures and Tables

**Figure 1 foods-11-01362-f001:**
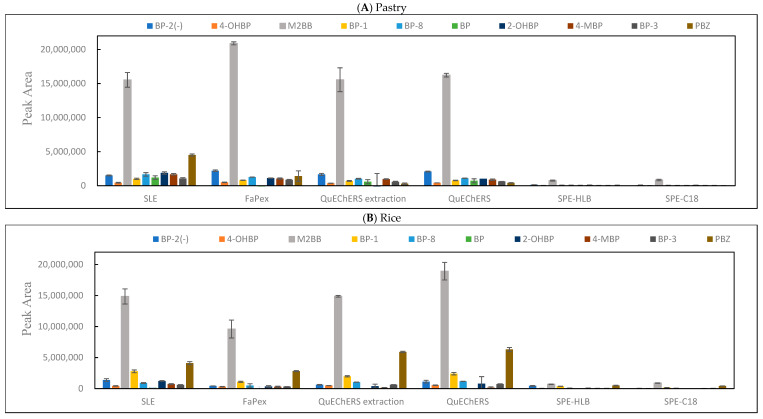
Comparisons of solid–liquid extraction (SLE), fast pesticide extraction (FaPEx), and quick, easy, cheap, effective, rugged, and safe (QuEChERS) using an extraction kit (4 g MgSO_4_ + 1 g NaCl), QuEChERS using an extraction kit (4 g MgSO_4_ + 1 g NaCl), and a cleanup procedure (0.24 g C18 + 0.24 g MgSO_4_ + 0.08 g PSA), SPE with HLB, and SPE with C18 in samples (*n* = 5 replicates) of (**A**) pastry, (**B**) rice, and (**C**) noodle spiked with standards and IS (20 and 8 ng/g).

**Figure 2 foods-11-01362-f002:**
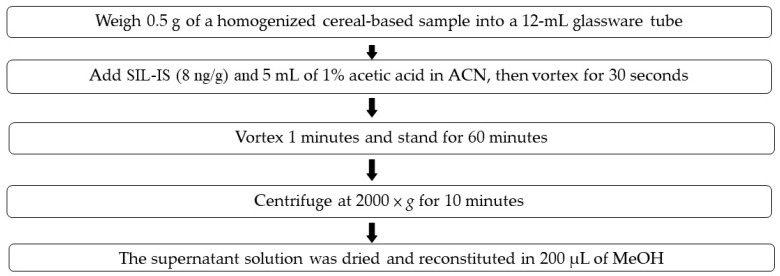
A flowchart of sample pretreatment, solid–liquid extraction.

**Figure 3 foods-11-01362-f003:**
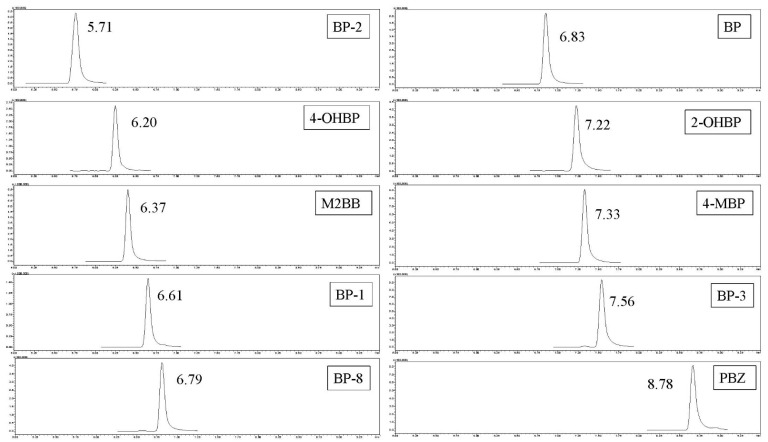
Chromatograms of a solvent-spiked sample of BP and BP analog standards at 80 ng/g.

**Table 1 foods-11-01362-t001:** Tandem mass spectrometry parameters for BP and BP analogs and corresponding stable isotope labeling (SIL).

Mass Spectrometer				Triple Quadrupole Shimadzu Tandem MS (Shimadzu-8045)			
Ionization				Electrospray Ionization			
Analyte Standards	MRM Iransition Ion (*m*/*z*)			Analyte SIL-Intenal Standards	MRM Transition Ion (*m*/*z*)		
	Precursor Ion	Quantitated Ion (CE1, V)	Qualified Ion (CE2, V)		Precursor Ion	Quantitated Ion (CE1, V)	Qualified Ion (CE2, V)
BP-2	[M − H]^−^ 245.0	135.0 (12)	109.0 (16)	d_4_-4OHBP	[M − H]^+^ 203.1	125.2 (23)	105.10 (23)
4-OHBP	[M − H]^+^ 199.0	121.1 (10)	77.1 (10)				
M2BB	[M − H]^+^ 240.3	209.1 (17)	152.0 (17)				
BP-1	[M − H]^+^ 214.9	137.0 (23)	105.0 (15)	d_5_-BP1	[M − H]^+^ 220.1	137.0 (25)	138.0 (25)
BP-8	[M − H]^+^ 245.0	121.1 (13)	151.0 (13)	d_3_-BP-8	[M − H]^+^ 248.1	121.1 (28)	154.1 (30)
BP	[M − H]^+^ 183.0	105.1 (19)	77.2 (19)	d5-BP	[M − H]^+^ 188.1	105.1 (21)	110.1 (22)
2-OHBP	[M − H]^+^ 199.2	121.0 (11)	93.0 (11)	^13^C_6_-di-OHBP	[M − H]^+^ 221.0	137.0 (11)	81.0 (24)
4-MBP	[M − H]^+^ 197.0	105.1 (21)	77.1 (10)	d_3_-4-MBP	[M − H]^+^ 200.2	105.1 (12)	77.1 (12)
BP-3	[M − H]^+^ 229.0	151.1 (25)	105.1 (11)	d_5_-BP_3_	[M − H]^+^ 234.1	151.0 (27)	81.9 (26)
PBZ	[M − H]^+^ 259.0	105.0 (10)	77.1 (29)				

**Table 2 foods-11-01362-t002:** Detection characteristics, linear range, matrix effect, limits of detection, and quantitation of target analytes in pastry.

Analytes	Pastry						
	Calibration Curve in Matrix-Matched	*R* ^2^	Calibration Curve in Solvent	*R* ^2^	Matrix Effect (%)	LOD (ng/g)	LOQ (ng/g)
BP	y = 0.0933x − 0.0439	0.998	y = 0.0963x − 0.1109	0.999	97	1.25	4.17
BP-1	y = 0.0585x − 0.0447	0.997	y = 0.0509x − 0.0321	0.999	115	0.11	0.36
BP-2	y = 0.1406x − 0.0584	0.997	y = 0.0898x + 0.0012	0.996	157	0.01	0.02
BP-3	y = 0.0651x + 0.0029	0.999	y = 0.0673x − 0.0377	0.998	97	0.23	0.77
BP-8	y = 0.0807x + 0.028	0.999	y = 0.0913x − 0.1022	0.999	88	0.20	0.65
2-OHBP	y = 0.0479x − 0.0016	0.996	y = 0.0569x − 0.0855	0.998	84	0.40	2.00
4-OHBP	y = 0.0897x − 0.0721	0.999	y = 0.1234x + 0.0056	0.997	73	0.08	0.26
M2BB	y = 1.2919x + 0.6299	0.996	y = 1.3811x − 0.2572	0.998	94	0.34	1.12
4-MBP	y = 0.0172x − 0.0011	0.999	y = 0.0172x − 0.0034	0.999	100	0.40	1.60
PBZ	y = 0.0419x − 0.1124	0.997	y = 0.0268x − 0.0251	0.996	156	0.20	0.40

**Table 3 foods-11-01362-t003:** Detection characteristics, linear range, matrix effect, limits of detection, and quantitation of target analytes in rice and noodle.

Analytes	Rice							Noodle						
	Calibration Curve in Matrix-Matched	*R* ^2^	Calibration Curve in Solvent	*R* ^2^	Matrix Effect (%)	LOD (ng/g)	LOQ (ng/g)	Calibration Curve in Matrix-Matched	*R* ^2^	Calibration Curve in Solvent	*R* ^2^	Matrix Effect (%)	LOD (ng/g)	LOQ (ng/g)
BP	y= 0.555 + 0.261x	0.999	y = 0.093 + 0.304x	0.999	82	0.04	0.4	y = 0.755 + 0.255x	0.996	y = −0.184 + 0.377x	0.999	115	0.04	0.4
BP-1	y = −0.050 + 0.169x	0.999	y = 0.066 + 0.152x	0.999	81	0.1	0.4	y = 0.460 + 0.166x	0.996	y = −0.060 + 0.172x	0.999	83	0.1	0.4
BP-2	y = 0.048 + 0.278x	0.999	y = −0.194 + 0.140x	0.995	60	0.01	0.02	y = 0.410 + 0.215x	0.997	y = 0.190 + 0.074x	0.996	26	0.01	0.02
BP-3	y = 0.062 + 0.217x	0.999	y = 0.077 + 0.212x	0.999	76	0.03	0.04	y = 0.391 + 0.192x	0.997	y = −0.045 + 0.275x	0.999	114	0.01	0.02
BP-8	y = 0.038 + 0.194x	0.999	y = 0.067 + 0.301x	0.999	96	0.04	0.3	y = 0.319 + 0.165x	0.996	y = 0.079 + 0.295x	0.998	138	0.03	0.3
2-OHBP	y = 0.011 + 0.077x	0.998	y = −0.006 + 0.116x	0.999	100	0.02	0.4	y = 0.429 + 0.090x	0.997	y = −0.096 + 0.092x	0.998	44	0.04	0.4
4-OHBP	y = 0.088 + 0.185x	0.999	y = 0.130 + 0.157x	0.998	81	0.3	1	y = 0.206 + 0.134x	0.999	y = −0.074 + 0.142x	0.995	103	0.3	1
M2BB	y = −0.068 + 0.737x	0.999	y = 1.223 + 0.524x	0.995	71	1	2	y = −0.748 + 0.498x	0.999	y = 0.087 + 0.488x	0.996	127	1	2
4-MBP	y = 0.008 + 0.090x	0.999	y = 0.044 + 0.073x	0.999	55	0.01	0.4	y = 0.021 + 0.192x	0.997	y = 0.250 + 0.191x	0.999	54	0.01	0.4
PBZ	y = −0.035 + 0.047x	0.998	y = −0.030 + 0.084x	0.999	106	0.03	0.3	y = 0.155 + 0.022x	0.996	y = −0.013 + 0.055x	0.998	113	0.1	0.4

**Table 4 foods-11-01362-t004:** Within-run and between-run recoveries in spiked pastry, rice, and noodle samples.

Analyte/ Spiked Levels (ng/g)	Pastry						Rice						Noodle					
	Within-Run (*n* = 5)			Between-Run (*n* = 15)			Within-Run (*n* = 5)			Between-Run (*n* = 15)			Within-Run (*n* = 5)			Between-Run (*n* = 15)		
	4	40	80	4	40	80	0.4	4	40	0.4	4	40	0.4	4	40	0.4	4	40
BP	88	110	90	86	117	90	84	86	98	87	94	97	114	104	104	106	100	102
BP-1	52	115	115	47	120	121	118	91	88	117	93	89	105	97	108	111	98	110
BP-2	120	134	148	111	130	142	89	71	76	89	70	80	119	118	115	117	113	115
BP-3	76	96	82	74	97	82	113	89	87	110	91	87	122	109	105	128	105	105
BP-8	58	89	73	58	90	75	122	91	88	123	92	87	122	108	97	130	112	92
2-OHBP	100	89	82	99	89	82	92	81	88	100	85	90	98	88	81	94	80	73
4-OHBP	131	72	73	134	73	73	94	95	104	98	94	101	118	115	87	126	118	77
M2BB	123	104	97	123	106	112	110	75	77	102	80	74	145	103	97	152	104	94
4-MBP	130	116	85	131	115	88	125	88	81	128	89	83	98	116	113	102	126	115
PBZ	146	51	45	150	50	44	111	104	75	109	115	73	98	88	75	106	86	96

**Table 5 foods-11-01362-t005:** Within-run and between-run precisions (%RSD) in spiked pastry, rice, and noodle samples.

Analyte/ Spiked Levels (ng/g)	Pastry						Rice						Noodle					
	Within-Run (*n* =5)			Between-Run (*n* = 15)			Within-Run (*n* = 5)			Between-Run (*n* = 15)			Within-Run (*n* = 5)			Between-Run (*n* =1 5)		
	4	40	80	4	40	80	0.4	4	40	0.4	4	40	0.4	4	40	0.4	4	40
BP	7	11	9	10	18	12	29	7	5	22	10	5	13	6	6	15	9	4
BP-1	4	2	3	6	5	7	3	2	5	7	2	4	8	3	3	10	5	3
BP-2	2	2	3	7	10	9	2	11	16	9	15	16	13	11	6	11	9	10
BP-3	2	4	2	5	6	4	1	3	3	5	4	2	7	1	1	6	6	1
BP-8	4	3	1	4	4	6	3	2	1	2	1	3	5	4	4	6	9	5
2-OHBP	5	2	3	9	11	13	10	8	14	11	9	18	5	9	19	8	13	18
4-OHBP	7	4	5	11	6	6	5	1	12	8	3	9	26	6	8	16	10	13
M2BB	4	6	5	7	8	16	12	20	16	19	20	18	8	7	3	7	9	6
4-MBP	3	2	3	5	3	6	2	10	7	4	6	7	3	5	2	7	21	2
PBZ	8	6	7	15	11	11	12	10	20	9	13	16	18	6	14	25	28	19

**Table 6 foods-11-01362-t006:** Summary of studies reporting the occurrence of BP and BPs in cereal-based foods and a comparison with the analytical methods.

Analytes	Country	Matrix	Analytical Methods	LOD/LOQ; Recovery (%)	Range (ng/g)	References
BP	UK	Food packaged in printed cardboard boxes, including cakes, burgers, rice	SLE with ACN and dichloromethane (DCM)–GC–MS	10/50 ng/mL	180–2000	Anderson et al. [19]
BP PBZ	Spain	Packaged baby food (cereals)	QuEChERS–HPLC–MS/MS	–/2.3 ng/g; Re = 97% –/0.7 ng/g; Re = 88%	Cereals: BP <LOD-40 PBZ: ND	Gallart et al. [22]
BP 4-MBP 4-OHBP	Germany	Cake, cookies cereals, couscous, noodles	SLE with ACN–HPLC–MS/MS	38/113 ng/g; Re = 94% 2.5/7.5 ng/g; Re = 89% 2.5/7.5 ng/g: Re = 86%	Cakes: BP:59, 4-MBP:13; noodles: BP:15; couscous: BP:867-1559	Jung et al. [23]
BP	UK	Cakes, cereals, sandwiches, burgers, snacks	SLE with ACN and DCM–GC–MS	NA	Cakes: <LOD-439	Bradley et al. [20]
BP	Switzerland	Cereal-based foods packaged in a cardboard box	PLE –GC–MS	–/60 ng/g; Re = 96–112%	5–7 × 10^6^	Bugey et al. [21]
BP 4-MBP	Belgium	Breadcrumbs, rice, pasta, cereals	SLE with ACN–UPLC–MS/MS	31/94 ng/g; Re = 87% 13/13 ng/g; Re = 92%	BP in breadcrumbs 5.2 and rice 3.6	Van Den Houwe et al. [24]
BP, 4-MBP 4-OHBP	Belgium	Cardboard-packaged dry foods (cereals, bread crumbs, pasta, rice)	SLE with ACN–SPE(HLB)–UPLC–MS/MS	BP, 4-MBP: 4/10 ng/g 4-OHBP: 0.1/0.6 ng/g	BP <4–20	Van Den Houwe et al. [25]
BP	Spain	Plastic-packaged foods (cakes, cookies, snacks)	SLE with ACN–GC–MS	10/250 ng/mL; Re = 85–115%	Cakes <10–54	Ibarra et al. [10]
9PIs: BP, 4-MBP, 2-OHBP	Taiwan	Cereals	QuEChERS–UPLC–MS/MS	LOD: 20, 10, 40 ng/g; Re = 84–123%	<LOD	Chang et al. [26]
BP and nine BPs	Taiwan	Cereals of oatmeal and corn flakes	FaPEx–UHPLC–MS/MS	0.001–0.3/0.03–0.8 ng/g; Re = 79–121%	BP:14–1084;4-MBP:1–66; BP-3: 0.1–8	Huang et al. [27]
BP and nine BPs	Taiwan	Pastries, rice, noodles	SLE with ACN–UHPLC–MS/MS	0.01–1.3/0.02–4.2 ng/g; Re = 44–150%, 70–128%, 73–152%	BP: 0.4–115; BP-1: 0.6–3; BP-3: 0.4–85; 2-OHBP: 0.5–25; 4-MBP:0.4–14.4	This study

**Table 7 foods-11-01362-t007:** BP and BPs levels in real samples (ng/g).

Analytes	Pastry (*n* = 25)			Rice (*n* = 50)			Noodle (*n* = 10)			*p*-Value ^a^
	DR (%)	Mean (SD)	Range	DR (%)	Mean (SD)	Range	DR (%)	Mean (SD)	Range	
BP	100	26.8 (32.6)	1.8–115.4	92	1.2 (2.0)	0.4–13.4	50	0.7 (0.7)	0.4–1.9	<0.0001
BP-1	76	1.6 (0.7)	0.6–2.7	2	0.9	–				0.67
BP-3	32	21.6 (37.6)	0.4–85.3							
2-OHBP	56	6.9 (6.4)	0.7–23.0	38	0.8 (3.6)	0.5–25.3				<0.0001
4-OHBP	4	1.5	–				10	1.4	–	
4-MBP	100	5.1 (4.2)	0.5–14.4	16	0.1 (0.2)	0.4–0.9				0.0001

DR: Detection rate; –: only one sample; ^a^ Mann–Whitney U test for BP and Kruskal–Wallis test for BP-1, 2-OHBP, and 4-MBP.

## Data Availability

All data and its Appendix A generated or analyzed during this study are included in this published article.

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
