# Peer review of "Development and Validation of Benzophenone Derivatives in Packaged Cereal-Based Foods by Solid–Liquid Extraction and Ultrahigh-Performance Liquid Chromatography–Tandem Mass Spectrometry"

_foods, 2022, doi:10.3390/foods11091362_

Round 1
Reviewer 1 Report
The present manuscript entitled "Development and validation of benzophenone derivatives in packaged cereal-based foods by solid-liquid extraction and ultrahigh-performance liquid chromatography–tandem mass spectrometry" by Yu-Fang Huang, Jun-Jie Huang, and Xuan-Rui Liu (foods-1688077) describes the development and validation of a novel and sensitive multi-residue analytical method for identifying benzophenone (BP) and nine BP derivatives. Ultra-high-performance liquid chromatography-tandem mass spectrometry and solid-liquid extraction pre-treatment were applied to the analysis of 85 packed cereal-based food samples from Taiwan (pastry, rice, and noodle samples). The proposed method is characterised by high sensitivity and specificity.
The present article is written correctly and has a good structure; moreover, it has all the necessary parts. The article is interesting from an analytical; therefore, it should interest the reader. My current decision is a minor revision. More specific comments and observations are presented below.
- Abstract. Some text is missing between lines 20 and 21. I am not sure if adding abbreviations to the abstract meets the journal's requirements. Please check it.
- RSD expressed as a percentage is the coefficient of variation (CV).
- Page 2, line 90. Literature references should be listed in ascending order. The same problem occurs elsewhere in the manuscript.
- Page 3, line 117. What were the parameters of the water used?
- What can be done in the event of strong interference effects? How would you deal with them? What types of interference effects could occur?
- Page 5, line 181. “pH” should be instead of “Ph”.
- Figure 1. The figure would look better in color.
- Tables 4 and 5. Titles with the number n are difficult to read. It is difficult to assign them to a specific column.
- The article can be diversified by adding the following elements: exemplary sample chromatograms, exemplary sample calibration plots, and a fragmentation diagram with formulas.
- Conclusions. Please, underline the differences in comparison with [27] and Liu et al., Foods 2022, 11, 572. Please, emphasize clearly the advantages of the research carried out.
- References. Please, adjust to the journal's requirements.
- The last empty page can be removed.
- Description of Fig. S1. Please add the "(B)".
I hope that the comments presented will help improve the article.
Author Response
- Abstract. Some text is missing between lines 20 and 21. I am not sure if adding abbreviations to the abstract meets the journal's requirements. Please check it.
Reply: Thank you for the comment. We have added “limits of detection” (line18) and deleted abbreviations.
2. RSD expressed as a percentage is the coefficient of variation (CV).
Reply: We thank the reviewer for this comment. We have revised the “coefficient of variation” in line 19.
3. Page 2, line 90. Literature references should be listed in ascending order. The same problem occurs elsewhere in the manuscript.
Reply: We thank the reviewer for this comment. We have revised the references list in ascending order (line 83) and throughout the manuscript.
4. Page 3, line 117. What were the parameters of the water used?
Reply: The Milli-Q water was produced by a Sartorius Ultrapure water system to reach a resistivity of 18.2 MΩ.cm. Please see line 110.
5. What can be done in the event of strong interference effects? How would you deal with them? What types of interference effects could occur?
Reply: LC–MS/MS is one of the most sensitive analytical methods, it often suffers from matrix effects. Matrix effects (MEs) can be observed either as an increase in response (ion enhancement) or as a loss in response (ion suppression). The ME was calculated as follows: (the slope of the matrix-matched calibration curve/the slope of the solvent calibration curve) × 100%. If ME=100%, there is no matrix effect. If ME> 100%, an ion suppression occurs, and, if ME <100%, ion enhancement occurs. The validation criteria of the Codex Alimentarius was 70%-120% for ME. The optimizing sample preparation procedures, manipulating LC and MS conditions, matrix-matched standards, and the use of stable isotopically labeled standards can compensate ME in this study.
6. Page 5, line 181. “pH” should be instead of “Ph”.
Reply: We thank the reviewer for this comment. We have revised it as “pH” in line 172.
7. Figure 1. The figure would look better in color.
Reply: We thank the reviewer for this comment. We have revised Figure 1 in color.
8. Tables 4 and 5. Titles with the number n are difficult to read. It is difficult to assign them to a specific column.
Reply: We thank the reviewer for this comment. We have revised it in Tables 4 and 5.
9. The article can be diversified by adding the following elements: exemplary sample chromatograms, exemplary sample calibration plots, and a fragmentation diagram with formulas.
Reply: We thank the reviewer for this comment. We have added “Figure 2. A flowchart of sample pretreatment, SLE” and “Figure 3. Chromatograms of a solvent sample spiked BP and BP analogs standards at 80 ng/g”.
10. Conclusions. Please, underline the differences in comparison with [27] and Liu et al., Foods 2022, 11, 572. Please, emphasize clearly the advantages of the research carried out.
Reply: We thank the reviewer for this comment. We developed and validated the FaPEx method in oatmeal and corn flakes [27] and rice cereal [35] samples. This is the first study in which a simple and efficient SLE UHPLC–MS/MS method was simultaneously applied for the identification of BP and nine BPs in samples of pastry, rice, and noodle. The method exhibited excellent results with the advantages of low background contamination. Specifically, LODs were below the nanogram per gram level, and analyses of BP and BPs had high recovery and precision in analyses of cereal-based foodstuffs. Please see lines 377-384.
11. References. Please, adjust to the journal's requirements.
Reply: We thank the reviewer for this comment. We have revised references.
12. The last empty page can be removed.
Reply: We thank the reviewer for this comment. We have removed it.
13. Description of Fig. S1. Please add the "(B)".
Reply: We thank the reviewer for this comment. We have added the "(B)".

Reviewer 2 Report
The manuscript ‘Development and validation of benzophenone derivatives in packaged cereal based foods by solid-liquid extraction and ultrahigh-performance liquid chromatography-tandem mass spectrometry’ has been reviewed and it’s a well-written manuscript. The authors are requested to consider the following suggestion for better clarity.
Suggestions:
- Authors may illustrate the manuscript concept in a flowchart format which may improve the presentation of the manuscript.
- Authors may include a brief comparison report of UHPLC method with other chromatographic method if possible.
- Authors are requested to include chromatogram figures of all BP standards
- The data of Table 2-Table 5 and Table 7 may present in graphical form instead of tables, which may improve the clarity of the manuscript.
- As the author has already published papers with the same concept which are mentioned in the reference section, what is the novelty of this research work.
Author Response
- Authors may illustrate the manuscript concept in a flowchart format which may improve the presentation of the manuscript.
Reply: We thank the reviewer for this constructive comment. We have added “Figure 2. A flowchart of sample pretreatment, SLE”.
2. Authors may include a brief comparison report of UHPLC method with other chromatographic method if possible.
Reply: We thank the reviewer for this constructive comment. UHPLC achieves a rapid chromatographic technique with a higher resolution, and lower mobile phase volume compared to HPLC. Please see lines 247-248.
3. Authors are requested to include chromatogram figures of all BP standards
Reply: We thank the reviewer for this comment. We have added “Figure 3. Chromatograms of a solvent sample spiked BP and BP analogs standards at 80 ng/g”. Please see lines 266-267.
4. The data of Table 2-Table 5 and Table 7 may present in graphical form instead of tables, which may improve the clarity of the manuscript.
Reply: We thank the reviewer for this comment. We keep the data and present it in Table format for data integrity.
5. As the author has already published papers with the same concept which are mentioned in the reference section, what is the novelty of this research work.
Reply: We thank the reviewer for this comment. The matrices in cereal-based foods are complicated and needed to be tested for suitability for sample pretreatment. We developed and validated the FaPEx method in oatmeal and corn flakes [27] and rice cereal [35] samples. We compared six pretreatment methods in samples of pastry, rice, and noodle and this is the first study in which a simple and efficient SLE UHPLC–MS/MS method was simultaneously applied for the identification of BP and nine BPs. The method exhibited excellent results with the advantages of low background contamination. Specifically, LODs were below the nanogram per gram level, and analyses of BP and BPs had high recovery and precision in analyses of cereal-based foodstuffs. Please see lines 377-384.
